# TORC1 regulation of dendrite regrowth after pruning is linked to actin and exocytosis

**Neeraja Sanal, Lorena Keding, Ulrike Gigengack, Esther Michalke, Sebastian Rumpf** *

Multiscale Imaging Center, University of Münster, Münster, Germany

* sebastian.rumpf@uni-muenster.de

## Abstract

Neurite pruning and regrowth are important mechanisms to adapt neural circuits to distinct developmental stages. Neurite regrowth after pruning often depends on differential regulation of growth signaling pathways, but their precise mechanisms of action during regrowth are unclear. Here, we show that the PI3K/TORC1 pathway is required for dendrite regrowth after pruning in Drosophila peripheral neurons during metamorphosis. TORC1 impinges on translation initiation, and our analysis of 5' untranslated regions (UTRs) of remodeling factor mRNAs linked to actin suggests that TOR selectively stimulates the translation of regrowth over pruning factors. Furthermore, we find that dendrite regrowth also requires the GTPase RalA and the exocyst complex as regulators of polarized secretion, and we provide evidence that this pathway is also regulated by TOR. We propose that TORC1 coordinates dendrite regrowth after pruning by coordinately stimulating the translation of regrowth factors involved in cytoskeletal regulation and secretion.

## Author summary

During development, neurons grow axons and dendrites that they use to make synaptic connections. Such connections are often fine-tuned through pruning and regrowth of axons and dendrites, but the coordination of the two processes is not well understood. It had previously been shown that hormone signaling suppresses the TORC1 growth pathway during pruning of Drosophila sensory neuron dendrites. We found that TORC1 is required for the subsequent regrowth of these dendrites. TORC1 activates protein biosynthesis, and our analyses suggest that it primarily targets neurite growth pathways, but not pruning pathways. These growth pathways include the actin cytoskeleton and the secretion machinery with the small GTPase RalA. Thus, the TORC1 growth pathway is a major hub coordinating neurite pruning and regrowth.

## Introduction

Developmental neurite remodeling is an important mechanism that serves to specify neuronal circuits, e. g., to adapt them to specific developmental stages. During remodeling, specific axons and dendrites can be removed through pruning, while others are maintained [1]. Cell

**Data Availability Statement:** All relevant data are within the manuscript and its Supporting Information files.

**Funding:** This work was supported by grant RU1673/6-1 from the Deutsche Forschungsgemeinschaft (DFG) and the

Collaborative Research Center CRC1348 (project B04) to SR. NS was supported by the Cells-in-Motion (EXC1003) graduate school and Cells-in-Motion bridging funds. The funders had no role in study design, data collection and analysis, decision to publish, or preparation of the manuscript.

**Competing interests:** The authors have declared that no competing interests exist.

biological pathways underlying neurite pruning therefore must involve mechanisms for spatial regulation of neurite destruction and maintenance. This spatial aspect of pruning may in part be due to cytoskeleton regulation [2] and tissue mechanical aspects [3]. Following pruning, remodeling neurons often regrow new neurites with a morphology adapted to subsequent developmental stages. The specific mechanisms underlying neurite regrowth after pruning are only beginning to be understood. In particular, it is interesting to ask how regrowth is coordinated with pruning at the signaling level, and if it uses different mechanisms than initial neurite growth.

Growth signaling pathways, and in particular the target of rapamycin complex 1 (TORC1) pathway have been implicated in axon regrowth after pruning [4,5]. TORC1 is a multi-subunit kinase that can be stimulated by diverse stimuli including amino acid sensing through the RAG complex, phosphatidylinositol-3-kinase (PI3K) signaling downstream of growth factor receptors [6], and through transcriptional activation of the TOR gene [5]. Interestingly, activation of TORC1 signaling also promotes axon regrowth after injury [7], suggesting that developmental regrowth pathways may be particularly well suited to stimulate growth also after injury. Thus, an understanding of regrowth pathways is also desirable in the medical context.

TORC1 mainly promotes growth by enhancing the rate of protein synthesis. This is achieved through either of two targets, RpS6 kinase (S6K) and eIF4E-binding protein (4E-BP) [6]. S6K is activated by TORC1 and promotes ribosome biogenesis. 4E-BP binds to and inhibits the translation initiation factor eIF4E, and TORC1-mediated phosphorylation inhibits it and thus derepresses eIF4E. Different mRNAs depend to varying degrees on eIF4E for translation initiation, and thus, TORC1 inhibition or activation has differential effects on the translatome [8–10]. The degree to which an mRNA depends on eIF4E for translation seems to be encoded in part in its 5' untranslated region (5' UTR). For example, ribosomal protein mRNAs often have short, so-called 5' terminal oligopyrimidine (TOP) motifs in their 5'UTRs that confer a strong dependence on eIF4E, and hence TORC1, for maximal translation. Other motifs governing TORC1 dependence also exist but are not well understood [9]. Whether TORC1 signaling confers any specificity during regrowth after pruning is not clear.

While several signaling pathways involved in axon regrowth after pruning are known, less is known for dendrite regrowth after pruning. In Drosophila, the peripheral class IV dendritic arborization neurons (c4da) prune and then regrow their sensory dendrites [11]. Recently, transcriptional [12] and posttranscriptional [13] gene regulatory pathways, specific cytoskeletal regulators [14], extracellular cues [15] and extracellular matrix remodeling [16] were shown to contribute to c4da neuron dendrite regrowth, but how they contribute to temporal regulation and specificity is still unclear. The TORC1 pathway has previously been shown to be downregulated during c4da neuron dendrite pruning [17], opening up the possibility of differential regulation. Furthermore, we recently provided evidence that translation initiation may be differentially regulated during dendrite pruning [18].

Here, we show that TORC1 is indeed required for dendrite regrowth after pruning. Experiments with 5' UTR reporter genes suggest that TORC1 selectively stimulates translation of regrowth, but not pruning factors linked to the actin cytoskeleton. Furthermore, we find that exocytosis via RalA and the exocyst is also required for dendrite regrowth, and we provide evidence that TORC1 is involved in the regulation of this pathway as well.

## Results

### TORC1 is required for c4da neuron dendrite regrowth after pruning

To visualize regrowth of c4da neuron dendrites after pruning, we first established a timecourse of regrowth. We chose to study the ventrolateral c4da neuron v'ada, because it remodels its

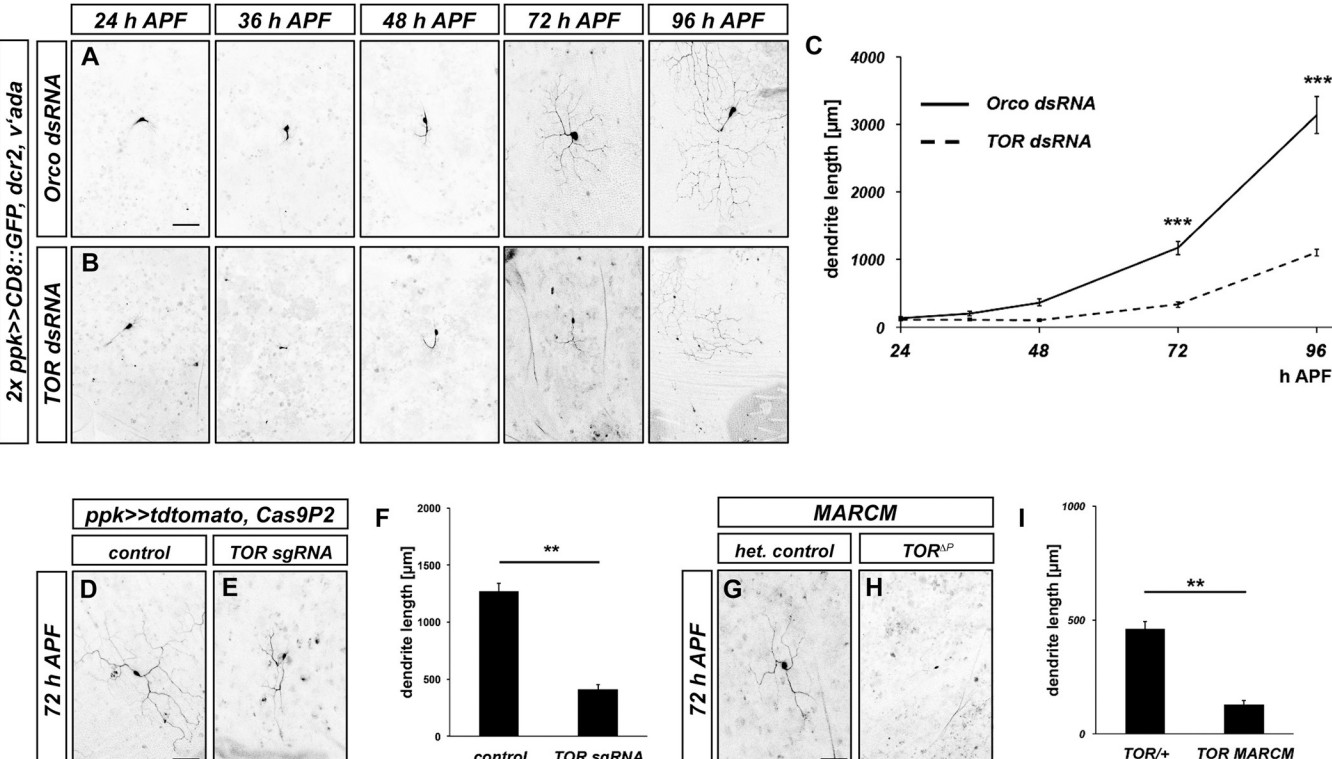

**Fig 1. TOR is required for v'ada neuron dendrite regrowth after pruning. A**, **B** Images show ventrolateral c4da neurons (v'ada) expressing the indicated dsRNA constructs under the control of *ppk-GAL4* at the indicated times in hours after puparium formation (h APF). **A** Regrowth timecourse of neurons expressing a control dsRNA construct against Orco. **B** Regrowth timecourse of neurons expressing TOR dsRNA. **C** Quantification of dendrite length over time. N = 9–12 neurons per genotype and timepoint. Values are mean +/- S.E.M., *** p<0.0005, two-way ANOVA test. **D** Cas9P2 expression under the control of *ppk-GAL4* at 72 h APF. **E** C4da neuron expressing Cas9P2 and two sgRNA constructs against TOR at 72 h APF. **F** Quantification of dendrite length in **D**, **E**. N = 12 neurons per genotype. Values are mean +/- S.E.M., ** p<0.005, Mann-Whitney U test. **G** C4da neuron heterozygous for *TOR^{ΔP}* and labeled with a *ppk-eGFP* promotor fusion. **H** C4da neuron MARCM clone homozygous for *TOR^{ΔP}* and labeled with *ppk-eGFP*. **I** Quantification of dendrite length in **G**, **H**. N = 11 neurons per genotype. Values are mean +/- S.E.M., ** p<0.005, Mann-Whitney U test. Scale bars in **A**, **D**, **G** are 50 μm.

dendrites with a similar timecourse as the dorsal c4da neuron ddaC [11,16], but is easier to visualize due to a lower number of autofluorescent bristles in its dendritic field. C4da neuron dendrite pruning is terminated at approximately 16–20 h APF [11,16]. We assessed v'ada neuron morphology at 24, 48, 72 and 96 h APF. As a control for dsRNA-mediated knockdowns, we expressed a dsRNA construct directed against the odorant coreceptor Orco, which is not expressed in these neurons, and does not affect dendrite regrowth (S1 Fig). In these control neurons, dendrite regrowth started shortly before 48 h APF (Fig 1A and 1C). Long branched dendrites had formed at 72 h APF, and full dendritic field coverage was reached at approximately 96 h APF (Fig 1A and 1C).

In a candidate screen for factors that affected v'ada dendrite regrowth, we identified TOR, a kinase involved in growth regulation. v'ada neurons expressing TOR dsRNA displayed a strong delay in dendrite regrowth, such that only a fraction of the dendritic field was covered at 72 h APF (Fig 1B and 1C). Consistent with the role of TOR in cell growth, the v'ada cell bodies also often appeared smaller upon TOR knockdown at this stage (Fig 1A and 1B). In order to confirm these results, we next used CRISPR/Cas9. Compared to control neurons only expressing Cas9, neurons expressing two sgRNAs targeting TOR had much shorter dendrites at 72 h APF (Fig 1D–1F). Furthermore, v'ada neuron MARCM clones homozygous for a TOR null mutation also displayed strongly reduced dendrite growth at this stage when compared to

heterozygous control neurons in the same animals (Fig 1G–1I, the seemingly shorter dendrite length in control animals in these experiments is due to the weaker fluorescence of the c4da neuron marker used to label v'ada).

To address whether TOR is also required for initial dendrite growth, we assessed the effects of TOR knockdown on elaboration of larval v'ada dendrites. Consistent with a previous report [19], loss of TOR led to a relatively mild—if any—decrease in v'ada dendrite length at the larval stage (S2 Fig). TOR knockdown had also no effect on dendrite pruning and expression of the pruning factor Mical in v'ada (S2 Fig), suggesting that this process was not affected by the loss of TOR.

TOR exists in two complexes, TORC1 and TORC2, that impinge on translation and the cytoskeleton, respectively. TORC2 has previously been shown to regulate dendritic tiling in larval c4da neurons [19]. To distinguish between these complexes during dendrite regrowth after pruning, we knocked down specific TORC1 and TORC2 subunits. Knockdown of Raptor, a TORC1 subunit, caused strong dendrite regrowth defects in v'ada, while knockdown of the TORC2 subunits Rictor and Sin1, or Tricornered (Trc) kinase, a TORC2 target in c4da neurons, only caused mild effects (S3 Fig). Furthermore, overexpression of a constitutively active form of the TORC1 downstream target S6K led to a significant rescue of the regrowth defects caused by TOR knockdown (S3 Fig). Thus, TOR, mostly via the TORC1 complex, is specifically required for dendrite regrowth after pruning.

## Dendrite regrowth requires the InR/PI3K/PDK1/Akt1 pathway

TORC1 can be activated by several upstream mechanisms, including the levels of specific amino acids and phosphatidylinositol-3 kinase (PI3K) signaling downstream of receptor tyrosine kinases [5]. In Drosophila mushroom body γ neurons, TOR has been reported to be activated by transcriptional upregulation through the transcription factors UNF and E75 [20]. In order to distinguish between these possibilities, we assessed the effects of dsRNAs or sgRNAs targeting these pathways on dendrite regrowth. Knockdown of UNF or E75 did not affect v'ada dendrite regrowth (Fig 2A–2D), and expression of dsRNAs targeting the RAG complex, which senses amino acids, also did not have an effect (Fig 2E–2H). In contrast, a dsRNA construct and an sgRNA targeting PI3K92E strongly reduced dendrite regrowth (Fig 2L, 2N and 2Q). Furthermore, dsRNAs against the phosphoinositide-dependent kinase PDK1 (Fig 2O and 2Q) and sgRNAs targeting the kinase Akt1 (Fig 2J, 2K and 2Q), which can both act downstream of PI3K, also strongly reduced dendrite regrowth.

The PI3K/PDK1/Akt1 pathway often acts downstream of receptor tyrosine kinases. As insulin signaling had been implicated in axon regrowth in Drosophila Crustacean Cardioactive Peptide (CCAP) neurons [4], we also tested the effect of downregulating the Drosophila insulin receptor InR. A dsRNA construct targeting InR caused a relatively mild reduction of v'ada dendrite regrowth (Fig 2P and 2Q), that appeared weaker than that of TOR, Akt1 or PI3K92E downregulation, leaving open the possibility that additional receptors could be involved in dendrite regrowth. Taken together, our data indicate that TOR regulation during c4da neuron dendrite regrowth involves PI3K/PDK1/Akt1 signaling downstream of the insulin receptor and possibly other receptors.

## An assay to assess TORC1 sensitivity of remodeling factor mRNAs

TORC1 promotes growth by upregulating translation through activation of S6K and inhibition of 4E-BP [6]. Specificity in this system arises through the differential sensitivity of various mRNAs to 4E-BP inhibition. As no TORC1 target mRNAs are known in the context of neuronal remodeling, we aimed to establish an assay for 4E-BP sensitivity of different remodeling

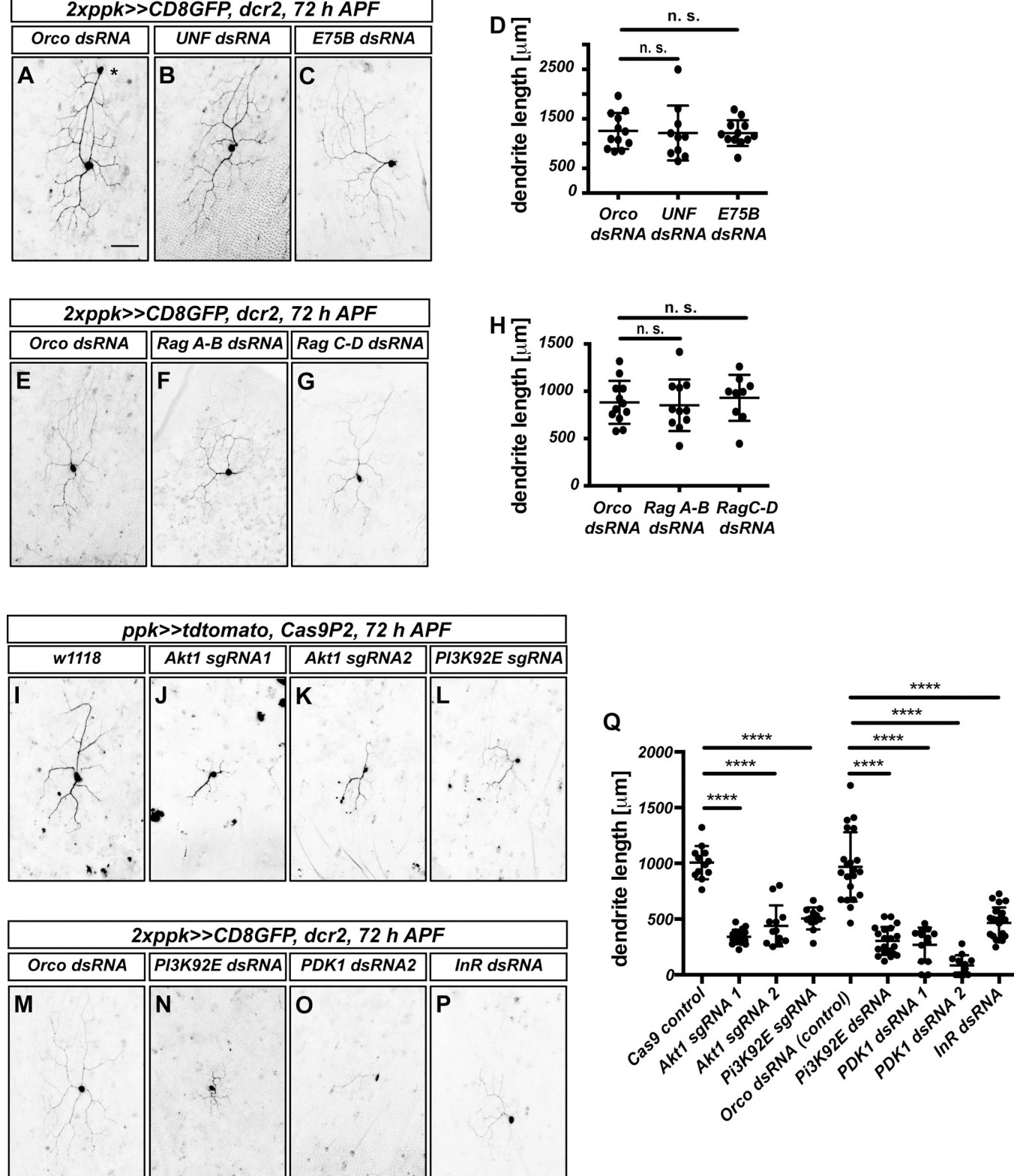

**Fig 2. The InR/PI3K/Akt pathway is required for v'ada neuron dendrite regrowth, but not the Rag complex or E75/UNF. A—D** UNF and E75 are not required for dendrite regrowth. Images show ventrolateral c4da neurons (v'ada) expressing the indicated dsRNA constructs at 72 h APF. **A** Orco control dsRNA. The asterisk denotes an unrelated ldaA-like neuron. **B** UNF dsRNA. **C** E75 dsRNA. **D** Quantification of dendrite length in **A—C**. N = 10–12 neurons

per genotype. Values are mean +/- S.D., n. s., not significant, Mann-Whitney U test. **E—H** The raG complex is not required for dendrite regrowth. **E** Orco control dsRNA. **F** Rag A-B dsRNA. **G** Rag C-D dsRNA. **H** Quantification of dendrite length in **E—G**. N = 10–12 neurons per genotype. Values are mean +/- S. D., n. s., not significant, Mann-Whitney U test. **I—P** Requirement for the InR/PI3K/Akt1 pathway during dendrite regrowth. Shown are ventrolateral c4da neurons (v'ada) expressing either Cas9P2 and the indicated sgRNA constructs or the indicated dsRNA constructs at 72 h APF. **I** Control neuron expressing Cas9P2. **J** Akt1 sgRNA #1. **K** Akt1 sgRNA #2. **L** PI3K92E sgRNA. **M** Orco control dsRNA. **N** PI3K92E dsRNA. **O** PDK1 dsRNA #1. **P** InR dsRNA. **Q** Quantification of dendrite lengths in **J—Q**. N = 12–21 neurons per genotype. Values are mean +/- S.D., **** p<0.0001, Mann-Whitney U test. The scale bar in A is 50 μm.

factor mRNAs. 4E-BP sensitivity is often encoded in the 5' UTRs of an mRNA. We therefore designed UAS transgenes driving expression of nuclear-localized, short-lived tdtomato [21] under the control of 5' UTRs from regrowth and pruning factor mRNAs (Fig 3A and 3B). To be able to account for effects of GAL4 strength, we additionally co-expressed UAS-CD8::GFP without a specific 5' UTR for normalization (Fig 3A). As ribosomal protein mRNAs are strongly regulated by TOR [8,10], we chose the 5' UTR of the Drosophila RpL13 mRNA as a bona fide candidate factor for a TOR/4E-BP target. Consistent with the importance for protein production during larval c4da neuron dendrite growth [22], RpL13 knockdown caused dendritic defects already at the larval stage and was strongly required for v'ada dendrite regrowth (S4 Fig). The small GTPase Rac1 had recently been shown to be required for v'ada dendrite regrowth via actin regulation [14]. As some mammalian actin isoforms are regulated by TOR [23] and the Drosophila actin isoform Act5C was also required for v'ada dendrite regrowth (S4 Fig), we chose the 5' UTRs of Rac1 and Act5C mRNAs as regrowth factor candidates. The actin severing enzyme Mical is required for ddaC neuron dendrite pruning [24]. We found that Mical is also required for v'ada dendrite pruning (S4 Fig), but not for v'ada dendrite regrowth (S4 Fig). We therefore chose the Mical mRNA 5' UTR as an example of a dedicated pruning factor (Fig 3B). We next expressed the 5'UTR reporters in larval ddaC c4da neurons and tested whether they were sensitive to overexpression of constitutively active 4E-BP (UAS-4E-BP LL). 4E-BP LL led to a threefold fluorescence reduction in the RpL13 reporter (Fig 3C–3C"). In support of the idea that regrowth factors are also regulated in a TORC1-dependent manner, the Act5C (Fig 3D–3D") and Rac1 reporters (Fig 3E–3E") showed an approximately twofold decrease of fluorescence. In contrast, the intensity of the Mical 5' UTR reporter was not significantly reduced (Fig 3F–3F"). Thus, our results are consistent with the hypothesis that translation of pruning and regrowth factors is differentially sensitive to 4E-BP activity and hence, regulation by TOR.

## TOR affects actin regulation during dendrite regrowth after pruning

The 5' UTR reporter assay indicated that TOR might be involved in actin regulation during dendrite regrowth. To test this idea, we expressed the F-actin marker lifeact::GFP in v'ada neurons and measured lifeact::GFP levels in the dendrites. To control for potential differences in reporter transcription levels, we normalized the lifeact::GFP levels to tdtomato. In control animals, lifeact::GFP could be readily detected in dendrites at both 48 and 72 h APF (Fig 4A, 4C, 4D and 4F). While the overall fluorescence intensity was lower in neurons lacking TOR, the lifeact::GFP/tdtomato ratio was similar to that of the control at 48 h APF (Fig 4B and 4C). In contrast, the lifeact::GFP/tdtomato ratio was significantly lower in the absence of TOR at 72 h APF (Fig 4E and 4F), indicating a reduction in dendritic F-actin during this period of dendrite regrowth. To make sure that the effect of TOR manipulation on dendritic actin was not due to a dendritic transport defect, we also assessed the expression of the endogenous c4da neuron ion channel Ppk26. In support of the idea that TOR loss does not cause a general transport defect, Ppk26 could still be detected in v'ada dendrites upon CRISPR-mediated TOR knockdown (S5 Fig).

**A**

**B**

| candidate 5'UTR | function |
|---|---|
| RpL13 | ribosomal protein |
| Act5C | actin isoform, regrowth |
| Rac1 | small GTPase, regrowth |
| Mical | actin severing enzyme, pruning |

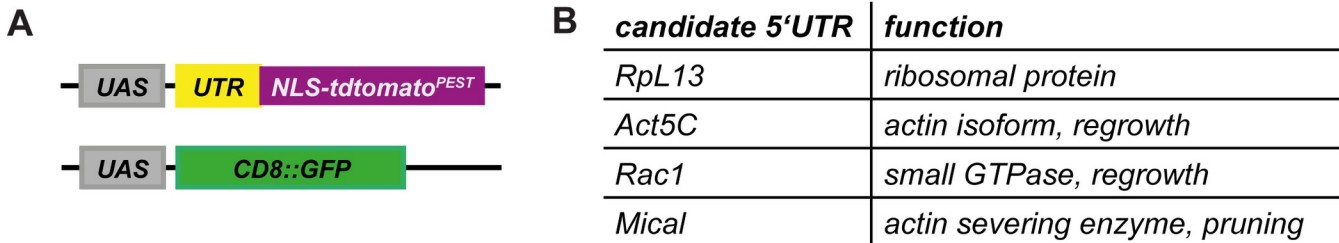

**Fig 3. 5'UTRs of remodeling factor mRNAs confer differential sensitivity to inhibition by 4E-BP in c4da neurons. A** Schematic depiction of 5'UTR reporter assay. Candidate 5' UTRs were inserted in front of the *NLS-tdtomato-PEST* reporter, and *CD8::GFP* expression was used for normalization. **B** Choice of mRNA candidates for 5'UTR analysis and their cellular functions. **C—F** 5' UTR reporter expression in control third instar c4da neurons. Left panels: single z slices (at the cell body position) from the tdtomato channel, right channels, projections of the tdtomato and GFP channels. **C'—F'** 5' UTR reporter expression in third instar c4da neurons upon coexpression of constitutively active 4E-BP LL. Left panels: single z slices (at the cell body position) from the tdtomato channel, right channels, projections of the tdtomato and GFP channels. **C"—F"** Quantification of tdtomato/GFP fluorescence intensity ratios in **C**—

**F'. C—C"** RpL13 5' UTR reporter (N = 10–11). **D—D"** Act5C 5' UTR reporter (N = 12–16). **E—E"** Rac1 5' UTR reporter (N = 20 per genotype). **F—F"** Mical 5' UTR reporter (N = 19–22). Values are mean +/- S.E.M., n. s., not significant, *** p<0.0005, Mann Whitney U test. The scale bar in **C** is 10 μm.

To further corroborate this effect of TOR on the actin cytoskeleton, we asked if TOR interacted genetically with Rac1 during v'ada dendrite regrowth. In control neurons, expression of GTPase-deficient, constitutively active Rac1$^{V12}$ did not lead to a significant increase in the total dendrite length at 72 h APF (Fig 4G, 4H and 4K). In contrast, Rac1$^{V12}$ caused a significant dendrite length increase in the TOR knockdown background when compared to TOR knockdown alone, indicating a partial rescue (Fig 4I–4K). Importantly, this rescue was not common to small GTPases in general, as expression of constitutively active Cdc42 did not lead to a

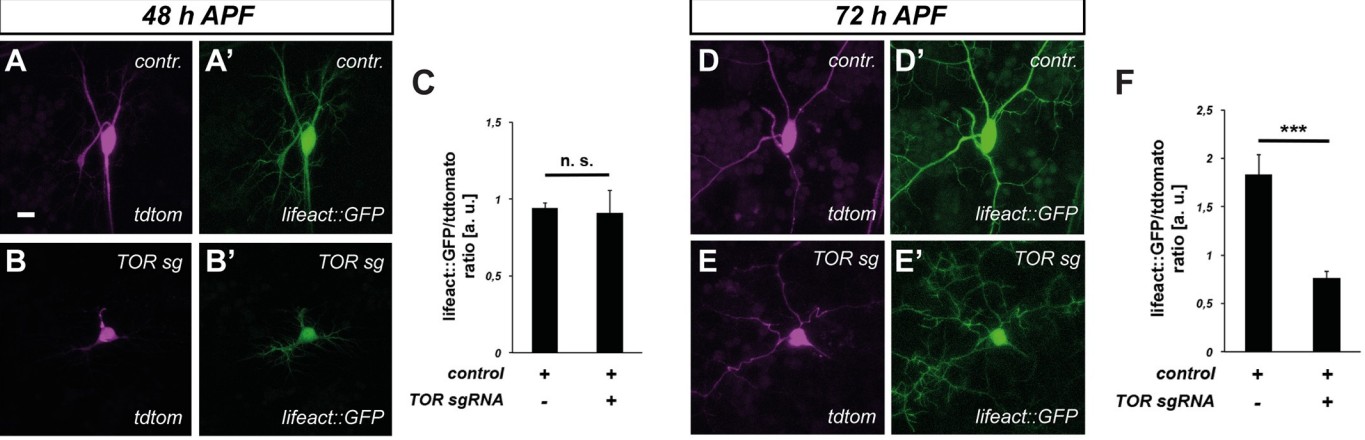

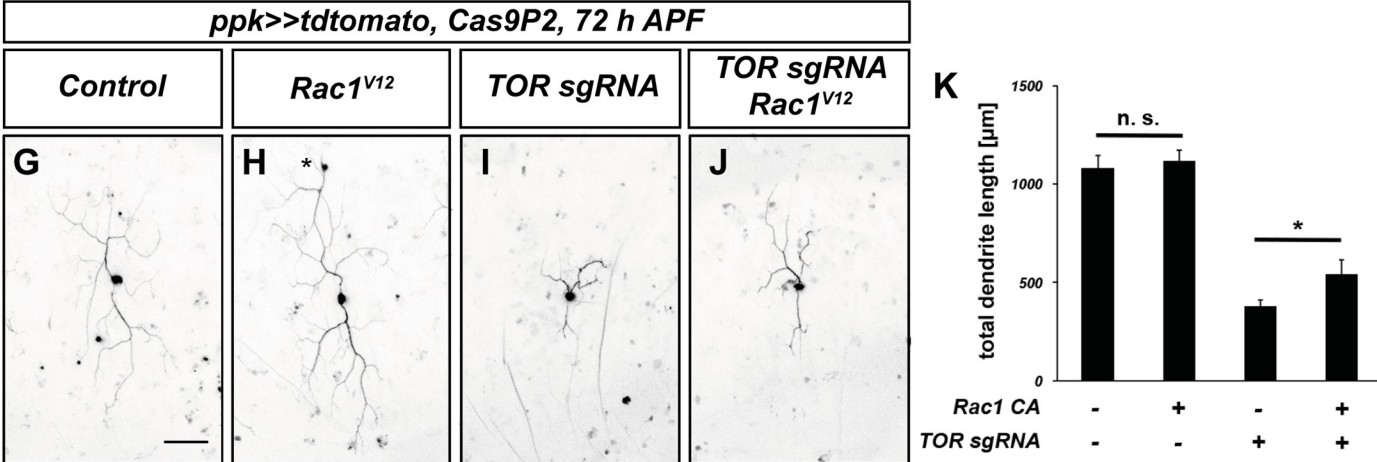

**Fig 4. Loss of TOR leads to lower F-actin levels during dendrite regrowth. A—F** F-Actin in c4da neurons was visualized using lifeAct::GFP, and neuronal morphology with tdtomato. **A, A'** Tdtomato and F-actin in a control c4da neuron at 48 h APF. **B, B'** Tdtomato and F-actin after TOR knockdown at 48 h APF. **C** Graph depicting normalized lifeAct::GFP intensity in dendrites in A—B' (N = 28, 12). Values are mean +/- S.E.M., n. s., not significant, Mann Whitney U test. **D, D'** Tdtomato and F-actin in a control c4da neuron at 72 h APF. **E, E'** Tdtomato and F-actin after TOR knockdown at 72 h APF. **F** Graph depicting normalized lifeAct::GFP intensity in dendrites in D—E' (N = 11, 12). Values are mean +/- S.E.M., *** p<0.0005, Mann Whitney U test. **G—K** Expression of constitutively active Rac1$^{V12}$ ameliorates regrowth defects caused by TOR knockdown. C4da neuron dendrite regrowth was assessed at 72 h APF. **G** Control c4da neuron labeled by tdtomato. **H** C4da neuron expressing Rac1$^{V12}$. The asterisk indicates a cell body of an ldaA-like neuron which sometimes is also labeled by *ppk-GAL4*. **I** C4da neuron expressing TOR sgRNA. **J** C4da neuron coexpressing TOR sgRNA and Rac1$^{V12}$. **K** Quantification of dendrite length in G—J (N = 11–14). Values are mean +/- S.E.M., * p<0.05, Mann Whitney U test. Scale bars are 10 μm in **A** and 50 μm in **G**.

rescue of the dendrite growth defects, even though Cdc42 was also required for v'ada neuron dendrite regrowth (S6 Fig). These data suggest that reduced actin polymerization via specific pathways contributes to the regrowth defects upon loss of TOR.

## The small GTPase RalA is required for dendrite regrowth after pruning

The above results raise the question whether TOR promotes dendrite regrowth by specifically promoting actin polymerization, or whether it also regulates other cellular pathways. To address this question, we sought to identify additional cellular processes required for regrowth. In our candidate screen for dendrite regrowth regulators, we identified the small GTPase RalA. RalA knockdown with two independent dsRNA constructs caused strong v'ada dendrite regrowth defects at 72 h APF (Fig 5A–5D). As with TOR, RalA knockdown did not cause apparent defects in larval v'ada neuron dendrite elaboration (S7 Fig), indicating that it might be specifically required at the regrowth stage.

Ral GTPases regulate membrane trafficking and various signaling pathways. Mammals possess two closely related Ral GTPases, RalA and RalB. Of these, RalB has been described as an upstream regulator of the TOR pathway [25], whereas RalA has been linked to membrane traffic. Drosophila only possesses the RalA gene, which has been linked to polarized secretion via the exocyst complex [26].

We next assessed RalA localization using a GFP-tagged RalA transgene [27]. At 48 h APF, GFP::RalA was evenly distributed in the soma and neurites of v'ada neurons (Fig 5E–5E"). In support of a role at the plasma membrane, GFP::RalA was clearly visible in higher order dendrites, indicative of plasma membrane localization, especially when compared to the cytosolic marker tdtomato (Fig 5F–5F"). In support of this notion, cytosolic tdtomato was mostly found in the center of a dendrite main branch, while GFP::RalA fluorescence clearly lined dendrite edges in fluorescence profiles from high resolution airy scan images (Fig 5G and 5H).

To test whether RalA was also linked to TOR signaling, we asked again if constitutively active, GTPase-deficient RalA$^{G20V}$ could rescue the dendrite regrowth defects induced by loss of TOR. v'ada neurons expressing RalA$^{G20V}$ alone did not have significantly longer dendrites than controls at 72 h APF (Fig 5I, 5J and 5M). In contrast, RalA$^{G20V}$ significantly ameliorated the dendrite regrowth defects in neurons expressing TOR sgRNA (Fig 5K–5M). Furthermore, RalA$^{G20V}$ expression also caused mild v'ada dendrite pruning defects during the early pupal stage (S7 Fig), similar to what had been described for TOR [17]. Thus, RalA might act in a TOR downstream pathway during dendrite regrowth.

## Polarized exocytosis is required for dendrite regrowth

As our data suggested that RalA might act at the plasma membrane, we next asked whether the exocyst complex could be involved in dendrite regrowth as well. This complex links exocytic vesicles to the plasma membrane via an interaction with Rab proteins and plasma membrane phospholipids [28]. It consists of the Sec5 subcomplex (containing Sec3, Sec5, Sec6, Sec10) and the Exo84 subcomplex (containing Exo70, Exo84, Sec8 and Sec15). Loss of the exocyst had previously been shown to cause defects in larval c4da dendrite branching [29]. We knocked down subunits from both subcomplexes using RNAi and found that loss of Sec6, Sec10, Exo70, Exo84 or Sec15 caused dendrite regrowth defects at 72 h APF compared to control neurons expressing Orco dsRNA (Fig 6A–6G). To verify these results with an independent method, we generated sgRNAs targeting Sec5 and Exo84 and co-expressed them with Cas9 in c4da neurons. Again, tissue-specific CRISPR knockdown of Sec5 and Exo84 caused strong v'ada regrowth defects at 72 h APF (Fig 6H–6K). Thus, polarized exocytosis via the exocyst is required for dendrite regrowth after pruning, strongly suggesting the involvement of the RalA-exocyst pathway.

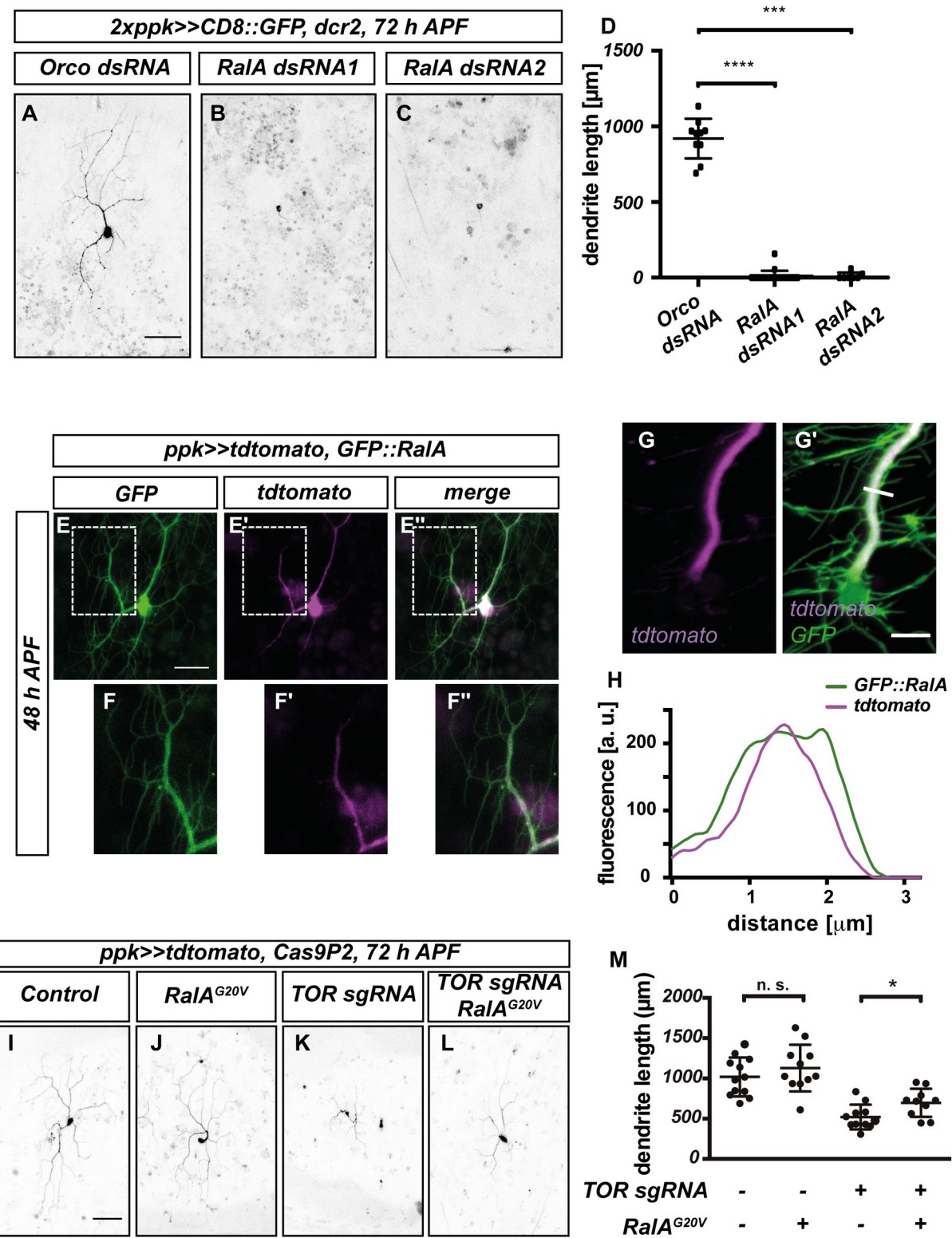

**Fig 5. RalA is required for dendrite regrowth and interacts genetically with TOR. A—D** Loss of RalA leads to dendrite regrowth defects. Images show c4da neurons expressing the indicated dsRNA constructs at 72 h APF. **A** Control c4da neuron expressing Orco dsRNA. **B** RalA dsRNA #1. **C** RalA dsRNA #2. **D** Quantification of dendrite length in A—C (N = 10–12 neurons per genotype). Values are mean +/- S.D., *** P<0.001, **** P<0.0001, Mann-Whitney-U test. **E—G** Localization of GFP::RalA in c4da neurons during dendrite regrowth. **E—E"** GFP::RalA localization pattern at 48 h APF. **E** GFP::RalA. **E'** tdtomato. **E"** Merge. Panels **F—F"** show the indicated regions in **E—E"** at higher magnification.

**G—H** Airy scan image of a v'ada dendrite expressing GFP::RalA at 48 h APF. **G** Tdtomato fluorescent signal. **G'** Merge with GFP:RalA signal. **H** Intensity profiles of tdtomato and GFP::RalA across the major dendrite branch at the position indicated by the white line in **G'**. **I—M** Genetic interactions between TOR and RalA during dendrite regrowth. C4da neurons of the indicated genotypes were labeled by tdtomato and imaged at 72 h APF. **I** Control neuron expressing Cas9P2. **J** Neuron expressing RalA$^{G20V}$. **K** Neuron expressing TOR sgRNA. **L** Neuron co-expressing TOR sgRNA and RalA$^{G20V}$. **M** Quantification of dendrite length in **I—L**. N = 11–12, values are mean +/- S.D., * P<0.05, Mann-Whitney-U test. Scale bars are 50 μm in **A**, **I**, 15 μm in **E**, and 5 μm in **G'**.

## Links between TOR and membrane regulation during dendrite regrowth

The above data opened up the possibility that exocytosis may be a downstream target of TOR signaling during c4da neuron dendrite regrowth. To test whether TOR manipulation affects membrane traffic during c4da neuron dendrite regrowth, we made use of a pHluorin-CD4-tdtomato reporter [30]. Here, an extracellularly exposed pHluorin moiety is linked to an intracellular tdtomato moiety via a transmembrane domain. While tdtomato fluorescence is unaffected by pH, the pHluorin moiety only fluoresces in a neutral pH environment, as is present on the cell surface, but not when in acidic environments like vesicles (Fig 7A). The pHluorin-CD4-tdtomato reporter can therefore be used to monitor endocytosis [31], exocytosis [29] and phagocytosis [30]. In control v'ada neurons at 48 h APF, pHluorin and tdtomato were clearly detectable in the growing dendrites (Fig 7B-7B" and 7D). tdtomato fluorescence intensity was comparable in dendrites of control v'ada neurons and neurons expressing TOR sgRNA. However, the pHluorin signal was significantly weaker in neurons expressing TOR sgRNA, resulting in a significant decrease of the pHluorin/tdtomato ratio (Fig 7C and 7D), suggesting that TOR downregulation may lead to defective or delayed membrane traffic during dendrite regrowth.

How could TOR be linked to exocytosis during dendrite regrowth? One possibility is that the 5' UTRs of membrane traffic factor mRNAs might also be sensitive to 4E-BP and disinhibition by TORC1. To test this idea, we generated a 5' UTR reporter using the 5'UTR of RalA mRNA. Indeed, the RalA 5' UTR rendered reporter expression partially sensitive to 4E-BP (Fig 7E–7E"). Thus, our results suggest that TOR promotes exocytic membrane traffic during dendrite regrowth through translational upregulation of membrane trafficking regulators.

## Discussion

Here we investigated a potential role and of TOR signaling in dendrite regrowth after pruning. PI3K/TOR signaling has previously been implicated in neurite (re-)growth, but both upstream signaling and its downstream targets have not been extensively explored. TORC1 had not previously been implicated in dendrite regrowth, but it had been shown that it must be downregulated during the pruning process preceding dendrite regrowth [17]. We found that the TORC1 complex is specifically required for dendrite regrowth after pruning in Drosophila sensory neurons, but not for initial growth of larval dendrites. We can envisage two possible explanations for this surprising specificity. For one, protein persistence and maternal contribution are known to mask early developmental phenotypes in Drosophila. However, the larval stage is a phase of intense growth and an abundance of nutrients. It is therefore interesting to speculate that another growth regulatory pathway may predominate—or be redundant with TOR—during the larval stage. For example, dMyc has been shown to regulate growth via transcription of ribosomal RNAs [32]. To better understand the role of TOR, we made an attempt to understand its target specificity. While the low number of c4da neurons per animal precludes a systematic biochemical identification of TOR target mRNAs (e. g., through ribosome profiling [8–10]), our results are consistent with a model where TORC1 broadly stimulates the translation of regrowth factors, including those linked to actin regulation and polarized exocytosis

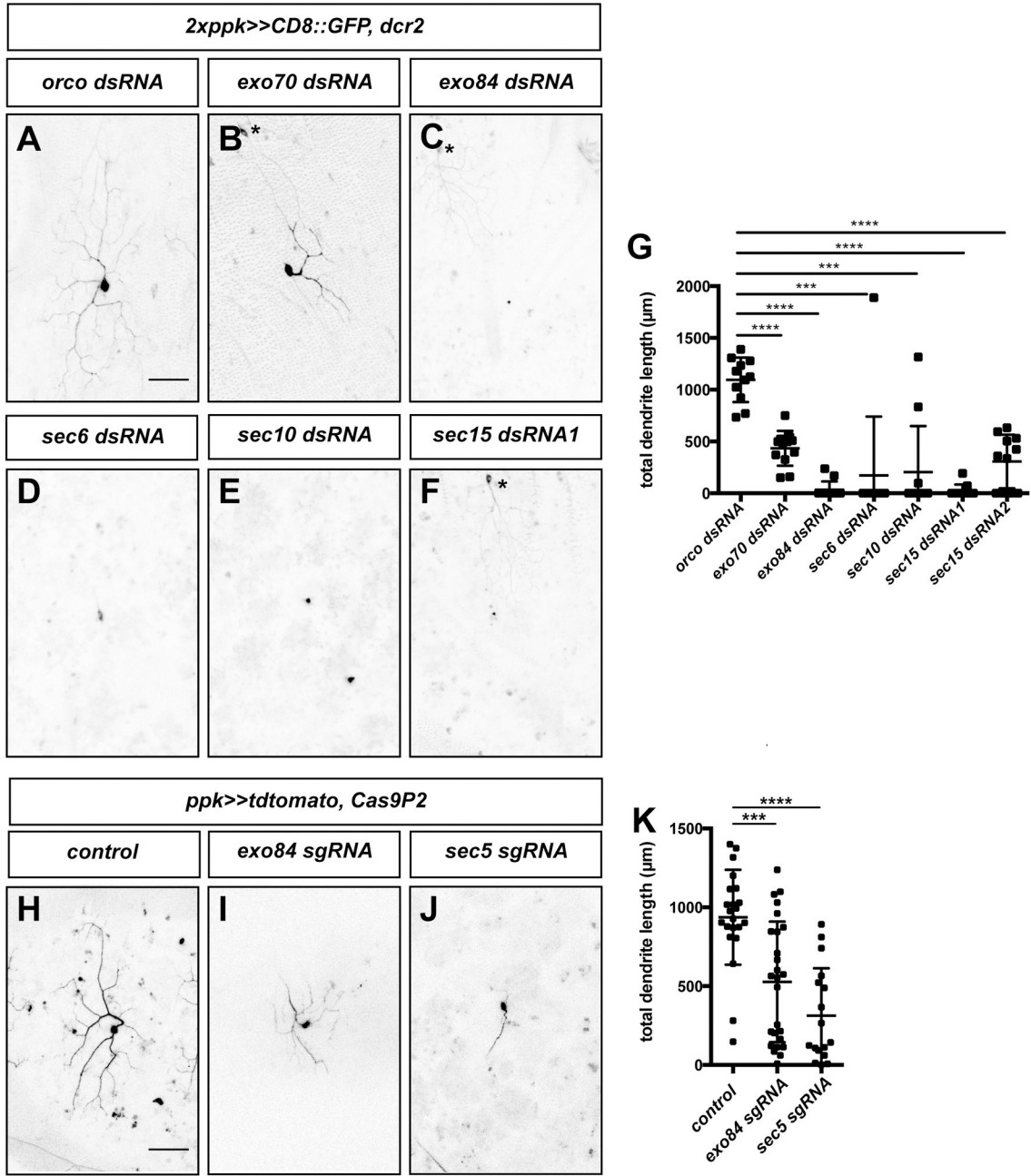

**Fig 6. The exocyst complex is required for dendrite regrowth.** Images show ventrolateral c4da neurons (v'ada) of the indicated genotypes at 72 h APF. **A—E** C4da neurons expressing dsRNA constructs under the control of *ppk-GAL4.* **A** Orco dsRNA. **B** Exo70 dsRNA. **C** Exo84 dsRNA. **D** Sec6 dsRNA. **E** Sec10 dsRNA. **F** Sec15 dsRNA #1. **G** Quantification of dendrite length in **A—F** (and including an additional independent Sec15 dsRNA2) (N = 10–12 neurons per genotype). *** P<0.001, **** P<0.0001, Mann-Whitney-U test. **H—J** C4da neurons coexpressing the indicated sgRNA constructs together with Cas9P2 under the control of *ppk-GAL4.* **H** Control without sgRNA. **I** Exo84 sgRNA. **J** Sec5 sgRNA. **K** Quantification of dendrite length in **H—J** (N = 18–27). Values are mean +/- S.D., *** P<0.001, **** P<0.0001, Mann-Whitney-U test. Asterisks indicate the positions of ldaA-like neurons co-labeled by *ppk-GAL4.* Scale bars in **A** and **H** are 50 μm.

(Fig 7F). The involvement of polarized secretion factors potentially points to a specific spatial regulation of dendrite regrowth, as the axons of c4da neurons are not pruned during the early pupal stage and therefore do not need to regrow [11]. In line with this idea, the exocyst has

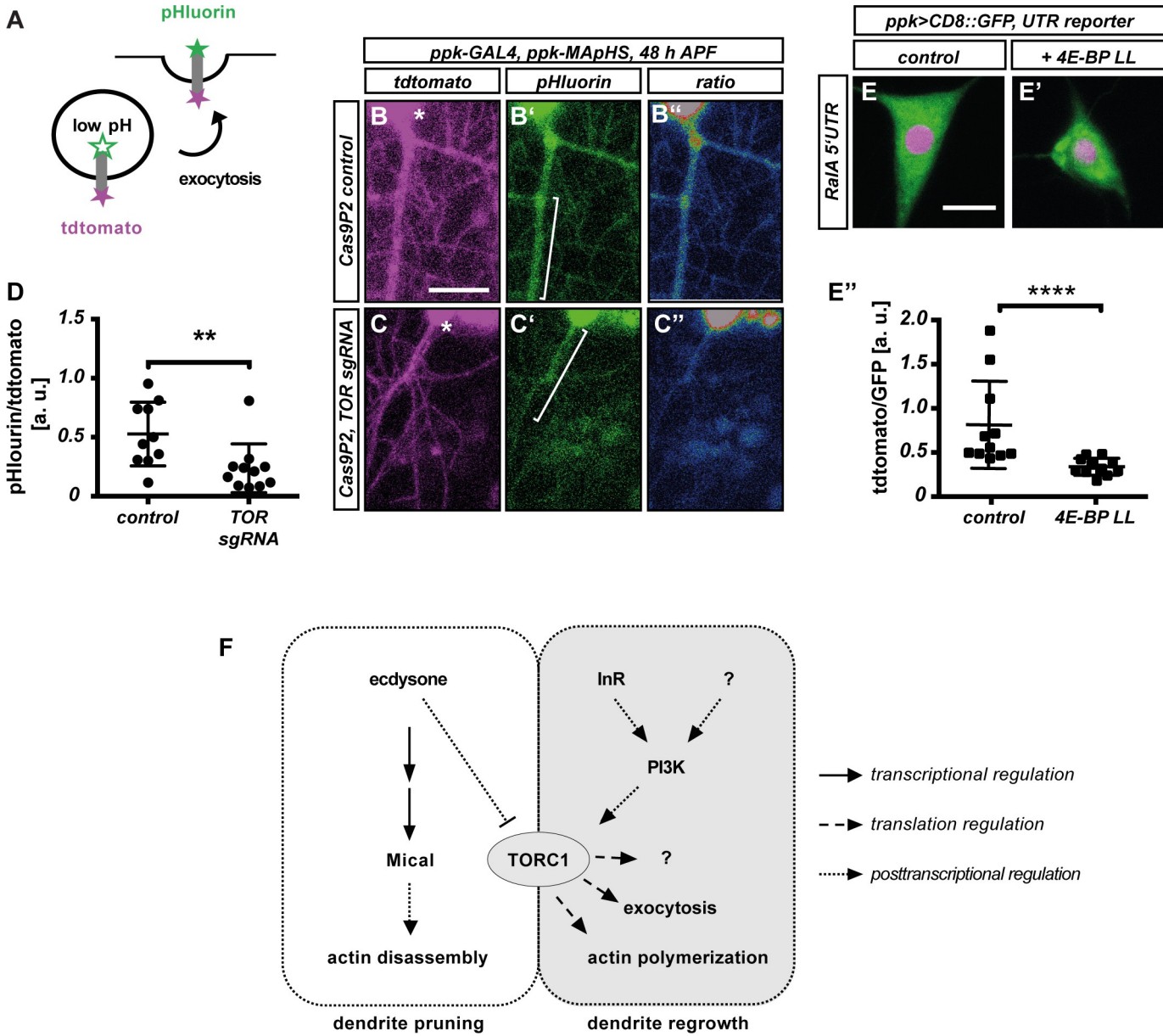

**Fig 7. Loss of TOR affects membrane trafficking during c4da neuron dendrite regrowth. A—D** Loss of TOR affects the distribution of the membrane trafficking marker pHluorin-CD4-tdtomato. **A** Schematic representation of MApHS design and readout. **B—C"** pHluorin-CD4-tdtomato fluorescence in c4da neuron dendrites at 48 h APF. Panels **B**, **C** show tdtomato fluorescence, **B'**, **C'** show pHluorin fluorescence, **B"**, **C"** show a pHluorin/tdtomato ratiometric image. **B—B"** Control neuron expressing Cas9P2, **C—C"** Neuron expressing Cas9P2/TOR sgRNA. Asterisks in B, C indicate the position of the soma. **D** Quantification of pHluorin/tdtomato ratio in primary dendrites at 48 h APF. Fluorescence intensities were measured in dendrite regions such as those indicated by brackets in **B**, **C**. N = 10–11, values are mean +/- S.D., ** P<0.005, Mann-Whitney-U test. **E** RalA 5' UTR reporter expression in control third instar c4da neurons. **E'** RalA 5' UTR reporter expression upon coexpression of constitutively active 4E-BP LL. **E"** Quantification of tdtomato/GFP fluorescence intensity ratios in **E**, **E'**. N = 10–11, values are mean +/- S.D., **** p<0.0001, Mann Whitney U test. **F** Hypothetical model for TORC1 regulation and function during c4da neuron dendrite pruning and regrowth. Ecdysone regulation of Mical and TOR is based on references 17, 24. Scale bars in **B**, **E** are 10 μm.

recently also been implicated in neurite regrowth after injury [33]. Despite the possibly very broad range of TORC1 targets, our analyses suggest that TORC1 also imposes specificity in that it does not promote the translation of the pruning factor Mical, and that its targets might have opposing functions to those regulated by the pruning trigger ecdysone (Fig 7F). We also

found that the effects of TOR loss can be partially suppressed by activated Rac1, but not by activated Cdc42, even though both GTPases are required for regrowth. We speculate that this difference may be due to a differential sensitivity of one of the Cdc42 and Rac1 downstream targets to TOR signaling. For example, during neuronal polarization in mammals, Rac1 regulates the WAVE complex [34], whereas Cdc42 regulates cofilin [35].

We recently found that the metabolically regulated AMP-dependent protein kinase (AMPK) is required for dendrite pruning in the fly PNS through adaptation of cellular metabolism [36]. AMPK is sometimes described as a TOR antagonist, and a recent study found that AMPK also antagonizes TOR during dendrite pruning [37]. It is therefore interesting to speculate that TOR may also act on neuronal metabolism during regrowth.

While a clear mechanism has been proposed that inhibits TOR during dendrite pruning in response to ecdysone [17], it is less clear what activates it during regrowth. Our results suggest that InR/PI3K/Akt1 signaling is upstream of TOR here, thus potentially linking dendrite regrowth to extrinsic hormonal cues. In order to better understand the temporal regulation of the process, it will be interesting to identify the ligand(s) involved. Overall, our results provide a rationale as to why TOR must be differentially regulated during neuronal remodeling and suggest pathways by which neuronal remodeling is linked to systemic developmental signals.

## Materials and methods

### Fly strains

All crosses were done at 25°C. For expression labeling of c4da neurons, we used *ppk-GAL4* insertions on the second and third chromosomes [38]. MARCM clones of the $TOR^{deltaP}$ mutant allele (BL 7014) were induced with *SOP-FLP* [39] and labeled by tdtomato expression under $nsyb-GAL4^{R57C10}$ [40] and a *ppk-eGFP* promotor fusion [41]. *ppk-MApHS* [30] was used to express pHluorin-CD4-tdtomato. UAS lines were UAS-tdtomato [30], UAS-Rac1$^{V12}$ (BL 6291), UAS-cdc42$^{V12}$ (BL 4854), UAS-RalA$^{G20V}$ (BL 81049), UAS-lifeact::GFP (BL 35544), UAS-4E-BP LL [42], UAS-GFP::RalA [27]. UAS-dsRNA lines were: TOR (BL 34639), raptor (#1: BL 31528, #2: 31529), rictor (#1: BL 31388, #2: BL 31527), sin1 (BL 36677), Trc (BL 28326), Act5C (VDRC 7139), RpL13 (VDRC 101369), Rac1 (VDRC 49246), RalA (#1: BL 29580, #2: VDRC 105296), Sec6 (VDRC 105836), Sec10 (BL 27483), Sec15 (#1: VDRC 105126, #2: VDRC 35161), Exo70 (VDRC 103717), Exo84 (VDRC 108650), Pdk1 (#1: BL 27725, #2: VDRC 109812), PI3K92E (VDRC 107390), InR (VDRC 992), E75 (BL 35780), Hr51 (BL 39032), Rag A-B (BL 34590), Rag C-D (BL 32342), Cdc42 (VDRC 100794). UAS-dsRNA lines were coexpressed with UAS-dcr2 [43], and Orco dsRNA (BL 31278) or mcherry dsRNA (BL 35785) were used as controls. Lines for CRISPR-mediated knockdowns were: PI3K92E sgRNA (BL 80898), Akt1 sgRNA #1 (BL 83097), Akt1 sgRNA #2 (BL 83502). UAS-Cas9P2 (BL 58986) was used for conditional CRISPR.

### Cloning and transgenes

5' UTRs of Act5C, Rac1, Mical, RpL13, and RalA were cloned in front of a short-lived tdtomato$^{PEST}$ (21) by PCR and inserted into pUAST attB via EcoRI and NotI sites. Transgenes were generated using the 86Fb insertion site. TOR sgRNAs TGACTCTCGTATAGCAGGTT (target 1) and CTTGGTGCTCTCTCGCTGAG (target 2) were cloned into pCFD3w+ (gift from S. Schirmeier) and injected into VK37 and attP2 landing sites, respectively. sgRNAs targeting Sec5 (TCCACTTGGGCTGTCCATCG (target 1) and TTCAAGCAGGAACCAGGCCG (target 2)) and Exo84 (TTTCAATGGCGTCCGCTAG (target 1) and CCGAAAGAAGTTGCCA CTAG (target 2)) were cloned into pCFD4w+ (gift from S. Schirmeier) and injected into flies carrying the 86Fb acceptor site.

## Dissection, microscopy and live imaging, immunostaining

For analysis of regrowth defects, appropriately staged pupae were dissected out of the pupal case at 24–96 h APF and v'ada in segments A2—A4 was visualized in live animals using a Zeiss LSM710 confocal microscope. All images were analyzed using Fiji [44], the NeuronJ plugin [45] was used for dendrite length measurements. NLS-tdtomato[PEST]/CD8::GFP intensity ratios of 5' UTR reporters were measured in the soma of third instar lateroventral (v'ada) c4da neurons. lifeact::GFP/tdtomato and pHluorin/tdtomato ratios were measured in primary dendrites at the indicated timepoints. Airy scan images of GFP::RalA were taken on a Zeiss LSM880 confocal microscope, and fluorescence profiles were generated in Fiji. Adult female abdominal bodywalls were dissected dorsally in PBS, fixed in 4% formaldehyde/PBS for 20 minutes, and blocked in goat serum. Filets were incubated overnight with rabbit anti-Ppk26 antibodies [46] and chicken anti-GFP (Aves labs cat. # GFP-1020). Mical stainings of early pupae were done as described [18]. Conjugated secondary antibodies were from Invitrogen.

## Quantification and statistical analysis

Dendrite regrowth phenotypes were analyzed by measuring dendrite length at the indicated timepoints using the Fiji NeuronJ plugin, these data were analyzed with Prism6 software using a two-tailed Mann-Whitney U test. Intensity ratios for NLS-tdtomato[PEST]/CD8::GFP (5' UTR reporters), dendritic lifeact::GFP/tdtomato and pHluorin/tdtomato ratios were also compared with a two-tailed Mann-Whitney U test.

## Supporting information

**S1 Text. List of fly genotypes.**
(DOCX)

**S1 Fig. Validation of Orco as control dsRNA for v'ada dendrite regrowth. A**, **B** Images show v'ada neurons expressing dsRNA constructs against mcherry (**A**) or Orco (**B**) under the control of two copies of *ppk-GAL4* at 72 h APF. **C** Quantification of dendrite length in **A**, **B**. N = 11 each, values are mean +/- S.D., n. s., not significant, Mann-Whitney-U test. The scale bar in **A** is 50 μm.
(TIF)

**S2 Fig. TOR is not required for larval c4da neuron dendrite growth or dendrite pruning. A**, **B** Images show v'ada neurons expressing Cas9P2 (**A**) or Cas9P2 and TOR sgRNA #2 (**B**) under the control of *ppk-GAL4* at the third instar larval stage. Neurons were visualized by tdtomato expression. **C** Quantification of dendrite length in **A**, **B**. N = 5 each, n. s., not significant, Mann-Whitney-U test. **D**, **E** Images show v'ada neurons expressing a control dsRNA construct against Orco (**D**) or TOR dsRNA (**E**) under the control of two copies of *ppk-GAL4* at the third instar larval stage. Neurons were visualized by CD8::GFP. **F** Quantification of dendrite length in **D**, **E**. N = 5 each, values are mean +/- S.D., n. s., not significant, Mann-Whitney-U test. **G**, **H** Images show v'ada c4da neurons Orco (**G**) or TOR dsRNA (**H**) at 18 h APF. **I** Percentage of neurons with unpruned dendrites in **G**, **H**. N = 20–31, n. s., not significant, Fisher's exact test. **J**, **K** Expression of the pruning factor Mical is not affected by TOR downregulation in v'ada. Mical was detected by immunofluorescence with the indicated antibodies at 2 h APF. **J**, **J'** Control v'ada neuron expressing Orco dsRNA. **K**, **K'** v'ada neuron expressing TOR dsRNA. Scale bars are 100 μm in **A**, **D**, 50 μm in **G** and 5 μm in **J**.
(TIF)

**S3 Fig. TORC1 is the main TOR complex required for c4da neuron dendrite regrowth after pruning. A—E** Images show v'ada c4da neurons expressing the indicated dsRNA constructs under the control of *ppk-GAL4* at 72 h APF. **A** Control dsRNA construct against Orco. **B** raptor dsRNA #1. **C** raptor dsRNA #2. **D** rictor dsRNA #1. **E** sin1 dsRNA. **F** rictor dsRNA #2, **G** Trc dsRNA. **H** Quantification of dendrite length in **A—G**. N = 12–20, values are mean +/- S.D., ** P<0.01, *** P<0.001, **** P<0.0001, Mann-Whitney-U test. **I—K** The loss of TOR can be compensated by constitutively active S6K during dendrite regrowth. **I** Neuron expressing TOR sgRNA. **J** Neuron co-expressing TOR sgRNA and S6K$^{STDETE}$. **K** Quantification of dendrite length in **I—J**. N = 8–11, values are mean +/- S.D., **** P<0.0005, Mann-Whitney-U test. Scale bars in **A** and **I** are 50 μm.
(TIF)

**S4 Fig. RpL13 and Act5C, but not Mical, are required for c4da neuron dendrite regrowth after pruning.** Images show v'ada c4da neurons expressing the indicated dsRNA constructs under the control of *ppk-GAL4* at the indicated timepoints. **A—C** Larval stage. **A** C4da neuron expressing a control dsRNA construct against Orco. **B** C4da neuron expressing RpL13 dsRNA. **C** C4da neuron expressing Act5C dsRNA. **D—F** C4da neurons at 72 h APF. **D** C4da neuron expressing Orco dsRNA. **E** C4da neuron expressing RpL13 dsRNA. **F** C4da neuron expressing Act5C dsRNA. **G** Quantification of dendritic field area in **D—F**. N is indicated in the graph. Values are mean +/- S.D., *** p<0.001, Mann-Whitney U test. **H—M** Mical is required for v'ada pruning, but not for dendrite regrowth. **H** v'ada neuron expressing Orco dsRNA at 18 h APF. **I** v'ada neuron expressing Mical dsRNA at 18 h APF. **J** Percentage of v'ada neurons with dendrite pruning defects in **H**, **I**. N = 25 and 27, respectively, values are mean +/- S.D., *** P<0.001, Fisher's exact test. **K** v'ada neuron expressing Orco dsRNA at 72 h APF. **L** v'ada neuron expressing Mical dsRNA at 72 h APF. **M** Quantification of dendrite length in **K**, **L**. N = 12 each, values are mean +/- S.D., n. s., not significant, Mann-Whitney U test. Scale bars in **A**, **D**, **H**, **K** are 50 μm.
(TIF)

**S5 Fig. Ppk26 dendritic localization in adult control and TOR knockdown neurons.** v'ada was labeled by CD8::GFP expression under *ppk-GAL4*, and GFP (**A**, **B**, green) and Ppk26 (**A'**, **B'**, magenta) were visualized by immunofluorescence in three-day old adult females. **A**, **A'** Control v'ada neuron expressing Cas9P2. **B**, **B'** v'ada neuron coexpressing Cas9P2 and TOR sgRNA. Note that the Z slices containing the cell body were omitted because of high background staining. The scale bar in **A** is 50 μm.
(TIF)

**S6 Fig. Cdc42 is required for c4da neuron dendrite regrowth, but does not interact with TOR.** Images show v'ada c4da neurons expressing the indicated transgenes at 72 h APF. **A** Control neuron expressing Orco dsRNA. **B** Neuron expressing Cdc42 dsRNA. **C** Quantification of dendrite length in **A**, **B**. N = 10–12, values are mean +/- S.D., *** p<0.001, Mann-Whitney U test. **D—F** Constitutively active Cdc42 does not rescue dendrite growth defects upon loss of TOR. **D** Neuron expressing TOR sgRNA. **E** Neuron co-expressing TOR sgRNA and Cdc42$^{V12}$. **F** Quantification of dendrite length in **E**, **F**. N = 12 each, values are mean +/- S.D., n. s., not significant, Mann-Whitney U test. The scale bar in **A** is 50 μm.
(TIF)

**S7 Fig. Effects of RalA manipulation on c4da neuron larval dendrite growth and dendrite pruning. A**, **B** RalA is not required for larval dendrite growth. Images show v'ada neurons expressing a control dsRNA construct against Orco (**A**) or RalA dsRNA (**B**) under the control of two copies of *ppk-GAL4* at the third instar larval stage. **C** Quantification of dendrite length

in **A**, **B**. N = 5 each, values are mean +/- S.D., n. s., not significant, Mann-Whitney-U test. **D**, **E** RalA activation causes dendrite pruning defects. Images show a control v'ada neuron (**D**) or a v'ada neuron expressing activated RalA$^{G20V}$ (**E**) under the control of two copies of *ppk-GAL4* at 18 h APF. **F** Quantification of phenotypic penetrance in **D**, **E**. N = 20 each, n. s., not significant, Fisher's exact test. **G** Lengths of unpruned dendrites in **D**, **E**. *, values are mean +/- S.D., P<0.05, Mann-Whitney-U test. Scale bars in **A**, **D** are 50 μm.
(TIF)

## Acknowledgments

We would like to thank C. Klämbt for support, M. Krahn for comments on the manuscript, A. Moore for helpful advice in the initial stages of this project and C. Han, C. Klämbt, S. Kadener, S. Schirmeier, M. Krahn and S. Noselli for sharing reagents. We would like to thank the Bloomington and VDRC stock centers for fly lines.

## Author Contributions

**Conceptualization:** Neeraja Sanal, Sebastian Rumpf.

**Data curation:** Neeraja Sanal, Sebastian Rumpf.

**Formal analysis:** Sebastian Rumpf.

**Funding acquisition:** Sebastian Rumpf.

**Investigation:** Neeraja Sanal, Lorena Keding, Ulrike Gigengack, Esther Michalke.

**Supervision:** Sebastian Rumpf.

**Writing – original draft:** Neeraja Sanal, Sebastian Rumpf.

**Writing – review & editing:** Sebastian Rumpf.

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
