## [Decision Letter · Decision Letter 0]

19 Dec 2022

Dear Dr Rumpf,

Thank you very much for submitting your Research Article entitled 'TORC1 regulation of dendrite regrowth after pruning is linked to actin and exocytosis' to PLOS Genetics. I apologize for the slight delay. Since my recent COVID infection, I have been having a fever and bedbound until today. <o:p></o:p>

The manuscript was fully evaluated at the editorial level and by independent peer reviewers. The reviewers all considered your paper to be potentially interesting, but all reviewers, especially Reviewer #1 and #2, raised some substantial concerns about the current manuscript. Based on the reviews, we will not be able to accept this version of the manuscript, but we would be happy to consider a much revised version that thoroughly address reviewers’ concerns with new experimental results.<o:p></o:p>

If you decide to revise the manuscript for further consideration at PLOS Genetics, please aim to resubmit within the next 60 days, unless it will take extra time to address the concerns of the reviewers, in which case we would appreciate an expected resubmission date by email to plosgenetics@plos.org.

Yours sincerely,

Yan Song, Ph.D.

Academic Editor

PLOS Genetics

Gregory P. Copenhaver

Editor-in-Chief

PLOS Genetics

Reviewer's Responses to Questions

**Comments to the Authors:**

**Reviewer #1**: Neurite pruning with or without re-extension is widely observed during the development of almost all animals, ranging from worm to human. Although the mechanisms involved in initiation and other aspects of neural pruning have been addressing extensively, it was recently the mechanisms of switching from pruning to re-extension was demonstrated. Currently, our knowledge on the molecular control of neurite re-extension temporal-wise and spatial-wise is limited. On the other aspect, neurite pruning and re-extension share some comment features with the pathological development after axon injury, though both events hold their own specific features. Drosophila c4da neuron is among a few model systems that has been widely used to address these questions. Unlike well studied ddac neurons, this paper focuses on the other C4 neuron, c4v’da. The author asked how TORC1 regulates neurite extension and identified two mechanisms. First, TORC1 selectively controls regrowth over pruning through its regulation on the 5’UTR of the mRNA of remodeling factors. Second, TORC1 controls neurite polarization through GTPase RalA. I found two key contributions this work may have. First, it showed the different mechanisms may be employed for the pruning and neurite neurite re-extension of v’ada than ddac. Second, how TORC1 complex may coordinate the growth and polarity of re-growing dendrites. Therefore, this work will expand our current understanding of neurite pruning and re-extension and will be attractive to certain population of audience. However, I do have some concerns (see below).

Major concerns

1. (page 6, line 4-5) “As c4da neuron dendrite pruning is terminated at approximately 16 h APF,...”. Do they mean all c4da neurons or specific the c4v’ada neurons? The reference should be cited. In addition, both “c4da neuron” and “c4v’ada neuron” were used to stand for c4v’ada neurons all over the manuscript. Yet “c4da neurons” can be different neurons than c4v’ada neurons. This cause confusion and need to be precise.

2. (Fig. 1 A-C) TOR dsRNA showed shorter dendrite length than control. Is this due to the delay of dendrite pruning processes thus the initiation of regrowth or just the delay of dendrite re-growth? The data shown in FigS1 showed eventually TOR dsRNA v’ada neurons underwent pruning like control but this did not address whether the initiation of pruning was delayed.

3. (Fig S2) TORC2 is required for the tiling of another c4da neuron, ddaC (Koike-Kumagai et al. 2009. EMBO J. 28, 3879-92). The authors found that knocking down the component of TORC2 has less effects on the neurite regrowth of v’ada neuron; in contrast, knocking down the component of TORC1 showed dendrite regrowth defect. This raises two interesting questions: (1) how similar the pruning of different C4da neurons is? (2) Whether knocking down TORC1 cause any defects in neurite growth and/or the tiling of v’ada? This needs to be clarified. Similar experiments as they did for RalA (Fig S5) is important.

4. (Fig 3) The regulation on Mical mRNA 5’UTR was used as an indicator of dendrite pruning. The author drew a strong conclusion that TOR complex differentially regulates the translation of pruning and regrowth factors (Page 9, line 17-19). Mical is required for the pruning of ddaC neurons; however, its involvement in v’ada neuron pruning is not clear. Because the results shown in Fig 1-2 and Fig S1-2 already suggested the pruning and neurite re-extension of different C4da neurons may have shared and specific mechanisms, the above conclusion is too strong.

5. (Fig. 5, Fig S5) Since the RalA RNAi phenotypes are very strong and RalA is specifically for neurite re-growth, it is to my surprise that the rescue effect of constitutively active form of RalA is so mild though significant. Did this manipulation cause any gain-of-function effects on the pruning or initial dendrite growth of v’ada? Do the authors have any thought on this?

6. In Fig 6, they nicely showed that knocking down or somatic knock-out some components of Sec5 and Exo84 subcomplexes lead to neurite-re-growth defects. However, these results cannot direct tell us whether they are involved in the membrane localization of RalA. Therefore, their conclusion (page. 12, line10-11) is overstated.

7. (Fig 7F) The schematic model is good. However, where the TORC1 is likely suggests it serves as the “switch” from pruning to neurite re-growth. This is not fully tested in the manuscript.

Minor comments

1. The titles of y and x axes are all in italic. This should be only for genotypes, otherwise it would cause some confusion.

2. (page 3, line 8-9) “.... it primarily targets neurite growth pathways, but not degenerative pathways.” At first glance, this was very confusing because pruning dose not equal to degeneration; they only share a subset of features and mechanisms.

3. They set an interesting question that whether neurite re-growth uses same or different mechanisms than initial neurite growth (page 4, line 11-13). However, when they visited this question in the result section, it was phrased as “To address whether TOR is generally required for dendrite growth,....” (Page 6, line 23-24). Such description is not easy for readers to link the result to the original question.

4. No scale bar was shown in Fig 1A and 1B. The soma of TOR dsRNA looked much smaller than control. Would also make the dendritic fields of RNAi cells look smaller than control accordingly? If Knocking-down TORC also affect the soma size, will this affect the quantification of dendrites in Fig 1C and those in following figures?

5. (Page 13, line 16) Typo. “We found that the TORC1 complex is specifically is required for dendrite regrowth......”

**Reviewer #2**: In the manuscript “TORC1 regulation of dendrite regrowth after pruning is linked to actin and exocytosis” Sanal et al reported a function for PI3K/TORC1 pathway in dendrite regrowth after pruning in Drosophila peripheral neurons during metamorphosis. They showed that TOR selectively stimulates the translation through 5’UTR of remodeling factor mRNAs linked to actin. They also showed that GTPase RalA and the

exocyst complex, which are regulators of polarised secretion, are controlled by TOR pathway. The similar function of TORC1 pathway in dendrite regrowth as axon regrowth is new, even though it has limited novelty. They provided some mechanisms explaining how TORC1 pathway regulates dendrite regrowth after pruning, which can be potentially interesting and important. However, several main conclusions were not well supported by the evidences. Therefore, I would like the authors to really address my critiques through additional experiments and/or argument in the text before publication.

1. The authors used Orco dsRNA as a control dsRNA under the assumption that Orco is not expressed in mechanosensory neurons. Is there any evidence for that? It was already shown that some mechanosensory ion channels are expressed in olfactory or other non-mechanosensory neurons (piezo for example was shown to be expressed in olfactory PNs from Xie et al, 2022). An ideal control dsRNA should be against some proteins that are absolutely absent from fly (Luciferase RNAi line is available form Bloomington center).

2. In Figure 3, the authors performed an interesting experiment showing that 4E-BP LL has a regulatory role on the 5’UTRs of RpL13, Act5C and Rac1 in cultured cells. However, they failed to show that the endogenous protein level is changed upon TOR knockdown in the mechanosesnory neurons. Even though they provided knocked down evidences for these genes in dendrite regrowth from wild type neurons, it does not prove RpL13L, Act5C and Rac1 is downstream of 4E-BP LL. Can the author detect some of these endogenous proteins in TOR knockdown neurons?

3. In Figure 4A-E it seems that TOR knock down not only changed lifeact/tdTomato ratio, but also dramatically downregulated tdTomato in the dendrites. This raised the possibility that the general protein transport machinery is disrupted by TOR knockdown, even though the lifeact may be affected to a more sever extend. Indeed the author showed later in Figure 6 and 7 that protein trafficking regulators are disrupted in TOR knockdown neurons. Can the authors clarify whether the F-actin polymerization is affected or dendritic protein trafficking? Maybe show examples of dendrite localized proteins that are not changed upon TOR knockdown?

4. In Figure 7 B-C without showing the expression level of pHluorin in the neurons, the authors could not conclude that the excocytosis is affected in TOR knockdown neurons.

**Reviewer #3**: The authors addressed the molecular mechanism of dendritic regrowth after pruning in Drosophila peripheral neurons during metamorphosis, and how it may coordinate with the underlying mechanism of dendritic pruning. Based on genetic analysis and some known information from the literatures, the authors demonstrate the contribution of the “PI3K/TORC1-4E BP-actin dynamics/membrane traffic” pathway. Overall, I find this study interesting and think that it could be potentially suitable for publication in PLOS Genetics. To help improve the manuscript, I would like to make the following suggestions.

Results in Figure 1 and Figure 2.

1. Based on panels B, E and H, the soma in all mutant images appeared to be smaller. This seems to suggest that the function of TOR is not specific to dendritic regrowth but to the overall growth of the cell. However, from fig. s1, TOR appears to have no effect on the sizes of soma and dendritic arbors at the larval stage. Do these observations suggest that c4da neurons adopt two different sets of molecules and mechanisms to control/maintain cell growths at different developmentnal stages? Could the authors provide analysis/discussions that reconcile these phenotypes in the text?

2. The authors state that “TOR, mostly via the TORC1 complex, is specifically required for dendrite regrowth after pruning.”. In this part, the authors used multiple types of mutant to demonstrate the contribution of TOR and two dsRNA lines for raptor. However, only one dsRNA line was used for rictor and sin1, respectively. Could the authors examine more mutant lines for rictor and sin1, just as what were done for TOR and raptor? In addition, the expression of S6K showed a significant rescuing effect, suggesting the contribution of the TORC1 pathway. Could the authors do a similar experiment to further rule out a role of the TORC2 pathway? These experiments would further strengthen the abovementioned conclusion.

3. No scale bars in panels A and B.

4. Why t-test for the data in panel C but U-test in panels F and I.

Results in Figure 3.

1. I concur with the authors about the logic of the results in this part, and the results are interesting. However, I think these results may simplify the facts and are limited in two aspects. First, it is not clear how the background expression and signaling network in larval c4da would affect the interpretation of the results. Second, unless a complete expression profiling was performed, one cannot be certain if the translation of pruning and regrowth factors are differentially regulated or to what extent is differentially regulated, namely that a few examples might not be enough. Therefore, I suggest that the authors carefully discuss the caveats and limitations of the analysis.

2. I suggest to add intensity calibration bars for all fluorescence images.

Results in Figure 4.

1. The different phenotypes on Rac1 and Cdc42 are interesting. It is however unfortunate that the authors did not further analyze the underlying mechanisms, and thus left this part a bit confusing. I suggest the authors should at least discuss the potential downstream regulators of actin dynamics that may account for the observations.

Results in Figure 5.

1. I feel a sudden logic transit in the beginning of this part because the motivation of jumping from actin dynamics to membrane traffic is not clear. In addition, it seems that there is no introduction on RalA. Given the overall logic of the work, I suggest to add a short paragraph before the results shown in Figure 4 to motivate the further study on actin dynamics and membrane trafficking, both as the representative downstream effectors of the TORC1 complex.

2. The membrane localization of RalA is not clear (Panel E and F). Could the authors try an imaging system with a better optical resolution to improve the quality of the data?

**Have all data underlying the figures and results presented in the manuscript been provided?**

Reviewer #1: Yes

Reviewer #2: Yes

Reviewer #3: Yes

PLOS authors have the option to publish the peer review history of their article (what does this mean?). If published, this will include your full peer review and any attached files.

Reviewer #1: No

Reviewer #2: No

Reviewer #3: No

---

## [Decision Letter · Decision Letter 1]

14 Apr 2023

Dear Dr Rumpf,

We are very pleased to inform you that your manuscript entitled "TORC1 regulation of dendrite regrowth after pruning is linked to actin and exocytosis" has been editorially accepted for publication in PLOS Genetics. Congratulations! 

As you will see, there are still some minor points raised by Reviewer #1, which we ask that your address when you prepare your final draft for the production team (the editorial team will not need to re-evaluate).

Thank you again for supporting open-access publishing. We are looking forward to publishing your exciting work in PLOS Genetics!

Yours sincerely,

Yan Song, Ph.D.

Academic Editor

PLOS Genetics

Gregory P. Copenhaver

Editor-in-Chief

PLOS Genetics

Comments from the reviewers (if applicable):

Reviewer's Responses to Questions

**Comments to the Authors:**

Reviewer #1: The authors have addressed most of my comments. The new results and revised statements do improve the manuscript. Although I have a couple of minor comments (see below), I support the publish of this manuscript in PLOS Genetics.

1. a table to list the genotypes of flies used in each figure.

2. (Line 22-23, Page 8) “TORC1 promotes growth by upregulating.........S6K and inhibition of 4E-BP”. Although related information was offered in the introduction section, citing references here to support this statement would be better.

3. The results of RNAi and control groups in Fig. S7F show similar degree of differences to those in Fig. S4J. However, no significant difference was detected between the two group in Fig. S7F. Why?

Reviewer #2: The authors have addressed most of my comments. I am supportive for publication of this manuscript in PLoS Genetics.

Reviewer #3: The revised manuscript has addressed all my questions that were raised in the first round of review. I am now supportive for the publication of this interesting study in Plos Genetics.

**Have all data underlying the figures and results presented in the manuscript been provided?**

Reviewer #1: Yes

Reviewer #2: Yes

Reviewer #3: Yes

PLOS authors have the option to publish the peer review history of their article (what does this mean?). If published, this will include your full peer review and any attached files.

Reviewer #1: No

Reviewer #2: No

Reviewer #3: No

**Data Deposition**

http://datadryad.org/submit?journalID=pgenetics&manu=PGENETICS-D-22-01306R1

**Press Queries**

---

## [Editor Report · Acceptance letter]

5 May 2023

PGENETICS-D-22-01306R1 

TORC1 regulation of dendrite regrowth after pruning is linked to actin and exocytosis 

Dear Dr Rumpf, 

We are pleased to inform you that your manuscript entitled "TORC1 regulation of dendrite regrowth after pruning is linked to actin and exocytosis" has been formally accepted for publication in PLOS Genetics! Your manuscript is now with our production department and you will be notified of the publication date in due course.

With kind regards,

Anita Estes

PLOS Genetics

On behalf of:
